# The Interaction of Food Allergy and Diabetes: Food Allergy Effects on Diabetic Mice by Intestinal Barrier Destruction and Glucagon-like Peptide 1 Reduction in Jejunum

**DOI:** 10.3390/foods11233758

**Published:** 2022-11-22

**Authors:** Yanjun Gu, Lu Yao, Tianyi Jiang, Huilian Che

**Affiliations:** Key Laboratory of Precision Nutrition and Food Quality, Key Laboratory of Functional Dairy, Ministry of Education, College of Food Science and Nutritional Engineering, China Agricultural University, Beijing 100083, China

**Keywords:** food allergy, diabetes, intestinal barrier, GLP-1

## Abstract

The increase in food allergies and diabetes leads to the assumption that they are related. This study aimed to (1) verify the interaction between food allergy and diabetes and (2) explore the potential mechanisms by which food allergy promotes diabetes. Female BALB/c mice were grouped into a control group (CK), an ovalbumin-sensitized group (OVA), a diabetes group (STZ), and a diabetic allergic group (STZ + OVA) (Mice were modeled diabetes with STZ first, then were given OVA to model food allergies), and an allergic diabetic group (OVA + STZ) (Mice were modeled food allergies with OVA first, then were given STZ to model diabetes). The results showed that OVA + STZ mice exhibited a more serious Th2 humoral response, and they were more susceptible to diabetes. Furthermore, when the OVA + STZ mice were in the sensitized state, the intestinal barrier function was severely impaired, and mast cell activation was promoted. Moreover, we found that the effect of food allergy on diabetes is related to the inhibition of GLP-1 secretion and the up-regulation of the PI3K/Akt/mTOR/NF-κB P65 signaling pathway in the jejunum. Overall, our results suggest that food allergies have interactions with diabetes, which sheds new light on the importance of food allergies in diabetes.

## 1. Introduction

Food allergy is an excessive response of the immune system to food allergens [1]. These allergies affect various organs, including the skin, digestive tract, and respiratory system [2]. Food allergies are reported to affect approximately 5% of infants and 3–4% of adults in Western countries [3]. The symptoms of food allergy vary from mild itching, diarrhea, and cough to severe life-threatening throat edema and anaphylactic shock [4]. Food allergies can cause chronic inflammation [5,6,7], and many studies have shown that inflammation can be linked to the pathogenesis of diabetes [8,9,10]. Diabetes occurs when islet β-Cells cannot secrete insulin or the body cannot use insulin efficiently [11] and its prevalence is increasing year by year [12], affecting approximately 8.3% of the global population [13]. Diabetes can be classified as type 1 diabetes (T1D), type 2 diabetes (T2D), maternal diabetes (MD), and other types of diabetes, with T2D accounting for 90–95% of diabetes [14]. Diabetes can lead to multiple complications, such as diabetic nephropathy [15], diabetic cardio-cerebrovascular complications, diabetic foot, and diabetic retinopathy [16]. The immune system is involved in the occurrence and development of these diseases and increases the overall risk of premature death [17].

Previous epidemiological studies have found that MD may increase the risk of allergic diseases, including food allergy, asthma, and atopic dermatitis [18,19,20]. Insulin resistance has also been shown to be associated with aeroallergen sensitization and allergic asthma, but not nonallergic asthma [21]. Children with cow’s milk allergy (CMA) may have an increased risk of T1D [22]. Moreover, Klamt and collaborators suggested that T1D was associated with a higher risk of the self-reported presence of immunoglobulin (Ig) E-mediated allergies [23]. However, a reduction in the frequency of allergic symptoms has been observed in children with T1D [24]. In addition, a previous study showed a higher prevalence of T2D in patients with atopic dermatitis (AD) than in the general population [25]. These findings suggest a possible interaction between diabetes and food allergies.

Food allergies and diabetes are associated with intestinal barrier function [26,27]. The intestinal barrier is one of the main defense mechanisms against harmful external substances, and the breakdown or dysfunction of this barrier is associated with local and systemic consequences [26]. During the effector phase of food allergies, mast cell threshing releases inflammatory mediators that increase intestinal permeability [28]. The destruction of mast cells and the transport of M cells aggravate food allergies [29]. Intestinal barrier leakage causes the translocation of bacterial products such as lipopolysaccharide, which induces an inflammatory response that becomes a risk factor for diabetes and intestinal barrier dysfunction [16].

Glucagon-like peptide-1 (GLP-1) is a hypoglycemic hormone derived from small intestinal L cells that increases the sensitivity of glucose to insulin-secreting beta cells [30]. It has been suggested to be a positive regulator of the gut-barrier function that can decrease inflammation and protect against intestinal damage. [31]. In that regard, both endogenous and exogenous GLP-1 have been shown to improve intestinal permeability and maintain intestinal barrier integrity [32,33]. GLP-1 receptor agonists are currently prescribed for diabetes [34]. Moreover, they have been found to reduce airway inflammation and airway mucus hypersecretion in allergic asthma patients by modulating interleukin signaling and inhibiting airway smooth muscle cell contraction [35,36]. A pilot observational cohort study has shown the efficacy of GLP-1 agonists to treat concomitant asthma in obese T2D patients [37]. mTOR in the intestine may link energy supply with the production of GLP-1 in L cells [38]. The proposed mechanism of action of GLP-1 is believed to occur through activation of the PI3K/AKT signaling pathway and downstream of the mechanistic target of rapamycin (mTOR) kinase [4,5,8]. GLP-1 protects insulin-secreting cells against apoptosis and restrains high glucose-induced apoptosis by impeding oxidative stress through PI3K-dependent signaling pathways [1,9]. The PI3K/AKT/mTOR pathway has been shown to play a key role in allergic diseases and diabetes [39,40,41]. In addition, the nuclear factor kappa B (NF-κB) is a transcription factor that mediates several immunological, inflammatory, and metabolic processes. Altogether, these observations suggest that GLP-1 could activate the PI3K/Akt/mTOR/NF-κB P65 pathway and act as a bridge between diabetes and food allergy.

To date, the interaction and mechanism between food allergies and diabetes have not been reported. Therefore, it is important to clarify whether food allergies enhance the occurrence of diabetes. There are different murine animal models to study food allergies and diabetes. Ovalbumin (OVA), the major egg white allergen, is a 45-kDa protein constituting about 54% of all egg white proteins [42]. However, OVA is not intrinsically immunogenic and therefore must be injected into the systemic circulation in the presence of adjuvants (substances that increase the immunogenicity of an antigen) [43,44]. Nevertheless, OVA is a T cell-dependent antigen that is commonly used as a model protein for studying antigen-specific immune responses such as food allergy or allergic asthma in mice. On the other hand, streptozotocin (STZ) inhibits insulin secretion and causes insulin-dependent diabetes mellitus due to its specific chemical properties, namely, alkylating potency [45]. STZ is commonly used to induce TID and T2D in mice. High-fat diet/streptozotocin-treated (HFD/STZ) animal model involves a combination of an HFD to bring about hyperinsulinemia, IR, and/or glucose intolerance followed by subsequent injection of STZ, which results in a severe reduction in functional β-cell mass. Altogether, these two stressors are designed to mimic the pathology of T2D [46]. HFD feeding followed by STZ injection can often be reported to be dyslipidemic, similar to the metabolic profile of type 2 diabetes in humans [47,48]. In mice, the combination of HFD and high-dose STZ is common for modeling T2D [49,50].

In this study, we investigated the interaction between food allergy and diabetes by delivering different combinations of OVA and STZ to mice. Furthermore, we explored potential mechanisms by which food allergies promote diabetes.

## 2. Materials and Methods

### 2.1. Animals

In this study, 4-week-old female BALB/c mice were purchased from Vital River laboratory in Beijing, China, and admitted to the special pathogen-free Animal Laboratory of the School of food science and Nutritional Engineering, China Agricultural University in Beijing, China (Reg.no. SCXK (Beijing)-2016-006). The animal room was kept at 22 ± 1 °C, 55 ± 5% humidity, 12 h of light/dark cycle, and 15 times/h air exchange. All animal experiments were carried out under the scheme approved by the animal experiment welfare and ethics Inspection Committee of China Agricultural University (No. 20193765) and in accordance with the ethical standard guidelines of China Agricultural University. All the efforts aim to minimize the pain of experimental animals.

### 2.2. Establishment of the Animal Model

To verify the potential mechanism between food allergy and diabetes, 4-week-old female BALB/c mice were selected to establish a model. There were five groups in this experiment. Female BALB/c mice were divided into the control group (CK), the OVA-sensitized group (OVA), the diabetes group (STZ), the diabetic allergic group (STZ + OVA), and the allergic diabetic group (OVA + STZ). Mice were fed adaptively for one week. The CK and OVA group were fed with normal diet (Standard diet, SD). The STZ, OVA + STZ, and STZ + OVA groups were fed high fat diet (HFD). SD and HFD were obtained from Beijing HFK Bioscience Co. (Beijing, China). In HFD (D12492), 60% of the calories are provided by fat (lard), 20% by protein, and the remaining 20% by carbohydrate. We used OVA (Sigma Aldrich, St. Louis, MO, USA) to build a food allergy mice model and used STZ (Sigma Aldrich, St. Louis, MO, USA) to build a diabetes mice model.

The twelve CK group mice were randomly selected by their body weight on day 0. Mice were given 100 μL of PBS on days 0, 7, 14, 21, 28, and 42 (Figure 1A and Figure 2A).

The twelve OVA group mice were randomly selected by their body weight on day 0. Mice were given 100 μL of normal saline dissolved 1 mg OVA and 10 μg cholera toxin (CT) (substances that increase the immunogenicity of OVA) [43,44] (Sigma Aldrich, St. Louis, MI, USA) adjuvant orally on days 0, 7, 14, 21, and 28, then mice were given 100 μL of normal saline dissolved 70 mg OVA intragastric administration on day 42 (Figure 1A).

The STZ group mice were given STZ solution with an injection volume of 10 mL/kg and a dose of 130 mg/kg after fasting for 12 h and fasting for 4 h after injection on day 28. After the injection, fasting blood glucose (FBG) of mice was monitored and twelve stable FBG at 11.1 mmol/L or above mice were chosen from 30 mice in the group [51] (Figure 2A).

The OVA + STZ group mice were given 100 μL of normal saline dissolved 1 mg OVA and 10 μgCT on day 0, 7, 14, 21, and 28. On day 30, STZ solution with an injection volume of 10 mL/kg and a dose of 130 mg/kg after fasting for 12 h, and fasting for 4 h after injection, then twelve stable FBG at 11.1 mmol/L or above mice were chosen from 30 mice in the group. On day 42, mice were given 70 mg OVA (Figure 2A).

The STZ+ OVA group mice were given the same dose of STZ (STZ solution with an injection volume of 10 mL/kg and a dose of 130 mg/kg after fasting for 12 h and fasting for 4 h after injection) on day 28, then twelve stable FBG at 11.1 mmol/L or above mice were chosen from 30 mice as samples. On days 30, 37, 44,51, and 58, mice 100 μL of normal saline dissolved 1 mg OVA and 10 μg CT. Then, mice were given 100 μL of normal saline dissolved 70 mg OVA intragastric administration on day 72 (Figure 1A).

Mice weight, food intake, and water intake were recorded once a week. The rectal temperature was measured by a digital thermometer at 0.5 h after the challenge. (At challenge stage, OVA enters the body again, it interacts with the IgE molecules on the target cells to mediate the bridging reaction, activating the downstream signal pathway, causing the target cells to degranulate and release the mediators, resulting in an allergic reaction).

The scoring criteria of allergic symptoms were recorded 30 min after the challenge. Allergic scores were subsequently graded following an adapted pre-established scale. More precisely, as follows: 0, no symptoms; 1, scratching at nose and head; 2, puffiness around eyes and mouth; 3, wheezing, labored respiration, display liquid diarrhea, cyanosis around mouth and tail; 4, no activity after stimulation or convulsion; 5, death [28].

CK, OVA, STZ, and OVA + STZ mice were sacrificed on day 42 and STZ + OVA mice were sacrificed on day 72. CK, OVA, STZ, and OVA + STZ groups just need 42 days to model successfully. Thus, CK, OVA, STZ, and OVA + STZ animals were sacrificed on day 42. According to our purpose, we just changed the model of the order of OVA in which food allergies occur in diabetes. To test whether diabetes could influence food allergy. We modeled diabetes first (These models with STZ and HFD need 30 days), then mice were given OVA and CT to model food allergies (STZ + OVA) (This model with OVA and CT need another 42 days) mice model, compared to the CK and OVA groups. Thus, the STZ + OVA model took 72 days to build, and mice were sacrificed on day 72. In this section, although the STZ + OVA group took much longer time to model than the other groups, we found that STZ treatment could not further disrupt the immune balance induced by OVA administration.

### 2.3. Pathological Section and Mast Cell Staining Analysis

After the mice were sacrificed, the jejunum, spleen, Visceral adipose (mesentery fat), pancreas, liver, and subcutaneous fat were removed and subsequently fixed in phosphate-buffered 10% formalin, and then embedded in paraffin blocks. A section from each paraffin block was stained with hematoxylin and eosin (H&E) to examine the pathologic structures of the tissues. At the same time, the mast cells of the jejunum of each group were stained with toluidine blue, and the number of mast cells and the degranulation reaction were observed. Images were obtained from fluorescence microscopy.

### 2.4. Glucose and Insulin Tolerance Test

Glucose and insulin tolerance tests were tested on day 41 in CK, OVA, STZ, and OVA + STZ groups and on day 71 in STZ + OVA group mice. For the glucose tolerance test, mice were fasted for 16 h and were given an intragastric administration of glucose (1.5 mg/g) in purified water. Blood samples were drawn from the tail before and 15, 30, 60, 90, and 120 min after the injection. For the insulin tolerance test, mice were fasted for 4 h and were given an intraperitoneal injection of insulin (1.0 U/kg) (11011018, Novo Nordisk, DEN). Blood samples were drawn from the tail before and 15, 30, 45, and 60 min after the injection. Blood glucose levels were determined using a glucometer (Abbott Laboratories, St. Louis, MI, USA).

### 2.5. ELISA

The blood of female mice in each group was collected 45 min after the last OVA intragastric administration (CK, OVA, STZ, and OVA + STZ mice were collected on day 42, and STZ + OVA mice were collected on day 72), and centrifuged at 4 °C, 5000 rpm/min to separate the plasma or serum, the stored at −20 °C. Plasma samples collected from orbital sinus were used to quantify OVA-specific IgE, IgG1, and IgG2a (reported out as OD450 nm mean ± standard) and histamine (ng/mL) by using commercial mouse ELISA kit (Neogen Corporation, Lexington, KY, USA). In the meantime, serum samples collected from the orbital sinus were used to quantify cytokines (IL-4, IL-5, IL-6 (pg/mL)), interferon (IFN)-γ (pg/mL), zonulin (ng/mL), GLP-1 (pg/mL), tumor necrosis factor (TNF)-α (pg/mL), and insulin (mIU/L) levels by using commercial mouse ELISA kit (Neogen Corporation, Lexington, KY, USA).

### 2.6. Real-Time PCR

Jejunum tissues in different groups were cut into small pieces and homogenized. The total RNA was extracted from jejunum by Trizol (Invitrogen Corp., Carlsbad, CA, USA) reagent, and reverse transcribed into cDNA by One-Step gDNA Removal and cDNA Synthesis Supermix kit (TransScript, Beijing, China). Quantitative real-time PCR analysis (qRT-PCR) was carried out with SuperReal PreMix Plus kit (SYBR Green, Shanghai, China) following the manufacturer’s protocol. All samples were tested in triplicate, and the relative quantification was calculated by 2^−ΔΔCt^ method, which calculated the relative expression changes in the target gene, normalized to the endogenous reference β-actin. The target genes including ZO-1, Claudin-1, occluding, mMCP-1, IL-6, SGLT1, GLUT2, PI3K, Akt, Mtor, and NF-κb p65 and their primer sequences are shown in Table 1.

### 2.7. Western Blot

Total proteins from the jejunum (50–70 mg) were isolated using a RIPA buffer supplemented with a protease inhibitor PMSF and quantified by the BCA protein assay kit (Dakewe, Shenzhen, China). Proteins were subjected to SDS-PAGE using a 10% acrylamide gel, the proteins were transferred onto polyvinylidene fluoride (PVDF) membranes (MilliporeSigma, Merck KGaA, Darmstadt, Germany), and the electroblotted membranes were incubated with specific primary antibody (PI3K (1:1000), Akt (1:1000), p-Akt (1:1000), mTOR (1:500) and NF-κB p65 (1:1000) were purchased from Abcam (Burlingame, CA, USA)) at 4 °C overnight. Immune complexes were detected using HRP-conjugated anti-rabbit IgG (1:1000) (Abcam Inc., Cambridge, MA, USA) and enhanced chemiluminescence (ECL) plus reagent (Pierce ECL Western blotting Substrate, Thermo Fisher Scientific Inc., Waltham, MA, USA). The films were scanned and quantified by Image J (NIH, Bethesda, MD, USA).

### 2.8. Statistical Analysis

All data from fasting glucose, AUC of GTT/ITT, the rectal temperature, the scoring criteria of allergic symptoms, leucocyte component distribution, ELISA, Real-time PCR, and Western Blot were expressed as mean ± standard error of the mean (SEM) and determined by Student’s t-test (unpaired, two-tailed). Comparison among the groups was analyzed statistically using one-way ANOVA followed by the Student–Newman–Keuls (SNK) multiple comparison test. A *p*-value of <0.05 was considered statistically significant.

The weight, food intake, and water intake of mice were expressed as mean ± SEM. The chi-square analyses were performed to compare the differences between the three groups in the success rate of animal experimental models. GraphPad Prism 5.01 was used for the graphical presentation.

‘Wu Kong’ platform (Omicsolution Company, Sichuan, China) was used for the graphical presentation of the proportion of leucocyte (Lymphocytes, monocytes, and granulocytes) component distribution.

## 3. Results

### 3.1. Establishment of Diabetic Allergic (STZ + OVA) Mice Model

To test whether diabetes could influence food allergy, as shown in Figure 1A, mice were modeled into the STZ + OVA group, compared to the CK and OVA groups. After being injected with STZ on day 28, we found that STZ + OVA mice showed reduced body weight (Figure 1B), accompanied by enhanced dietary intake (Figure 1C) and water consumption (Figure 1D). Importantly, the fasting glucose levels significantly increased in the STZ + OVA group mice (Figure 1E). Furthermore, STZ + OVA mice also showed deteriorated metabolic fitness (Figure 1F,G) and significantly decreased insulin sensitivity (Figure 1H,I) compared to CK and OVA mice, indicating that STZ significantly weakened the glucose sensitivity of mice. Detailed tissue analysis revealed significant pathological alterations in the structure of the spleen (the lymphocyte of the red pulp area decreased and the area of the white pulp area increased) and adipocytes (the adipose cells were not tightly packed and well-defined) in STZ + OVA mice (Figure 1J,K). STZ + OVA mice were also compared to STZ mice. There was no significant difference between STZ and STZ +OVA mice in fasting glucose, and insulin sensitivity (Appendix A). Altogether, these data suggested that the diabetic allergic model was successfully developed.

### 3.2. Effects of Diabetes on Food Allergy

We next investigated whether diabetes could affect some of the immunological indicators associated with food allergies. Critically, rectal temperature was significantly altered in the OVA and STZ + OVA groups, while significantly allergic symptoms elevated were observed in those mice (Figure 3A,D). Strikingly, both the OVA and STZ + OVA groups showed significant alterations in antibody levels (IgE, IgG1, IgG2a, and IgG1/G2a), allergic cytokines (IL-4/5), IFN-γ, and histamine compared to the CK mice (Figure 3B,C,E,F). Furthermore, there were no significant differences observed between OVA mice and STZ + OVA mice with respect to the immunological changes, suggesting that OVA treatment significantly altered the immunological response, while STZ administration did not aggravate this phenotype in the STZ + OVA group (Figure 3B,C,E,F). Overall, these results indicated that STZ treatment could not further disrupt the immune balance induced by OVA administration.

### 3.3. Establishment of Allergic Diabetic (OVA + STZ) Mice Model

To study the impact of food allergy on diabetes, as shown in Figure 2A, we modeled the OVA + STZ group and compared it to the CK and STZ groups. First, after injection with STZ on day 28, we found a significant decrease in body weight and an increase in food and water intake in the STZ and OVA + STZ groups compared to the CK mice (Figure 2B–D). Compared to the CK group, mice of the STZ and OVA + STZ groups had inflammatory cell infiltration in the pancreas, were swollen and arranged as loosely scattered with the white vacuole formed by different lipid droplets in hepatocytes, and had increased subcutaneous fat cell hypertrophy (Figure 2E). Furthermore, fasting glucose levels in the STZ and OVA + STZ groups were significantly higher than those in the CK group (Figure 2F). In addition, the OVA + STZ group showed a decrease in metabolic fitness and insulin sensitivity compared to the CK mice (Figure 2G–J). These data supported the successful development of the allergic diabetes model and indicated that food allergies did not further disrupt glucose dysregulation in mice.

### 3.4. Effects of Food Allergy on Diabetes

To further study the influence of food allergies on diabetes, we evaluated the changes in leukocyte subsets and blood biochemical levels in the three groups of mice. The assessment of leukocyte subsets showed that the proportion of neutrophils increased significantly and that of lymphocytes decreased significantly in the STZ and OVA + STZ groups, with the decrease being more evident in the OVA + STZ group (Figure 4A and Appendix A). We also observed an increase in LDL-c in the OVA + STZ group compared to the STZ group (Figure 4B). Altogether, these results indicated that the pathological morphology of the viscera, composition of peripheral blood leukocytes, and metabolic level were altered when the body is sensitized to OVA.

We then set out to investigate if food allergies could affect diabetes by analyzing immunological indicators. We first analyzed the rectal temperature and food allergy symptom scores of mice after intragastric administration of high-dose OVA. Rectal temperature and food allergy symptom scores of the OVA + STZ group showed a significant increase compared to those of the control group, which indicated mice in the OVA + STZ group experienced more serious food allergic reactions (Figure 4C,D). Significant immunological alterations were found in the OVA + STZ mice, including reduced IFN-γ and IgG1/G2a and improved specific antibody (IgE, IgG1, IgG2a), IL-4/5, and histamine levels (Figure 4E–H). Importantly, we found that OVA + STZ mice showed significant alterations (IL-4 and IL-5 increased significantly, whereas a decrease in IFN-γ was observed significantly) in the immunological indicators compared to the STZ group, indicating that the immune response of OVA + STZ mice was significantly inclined toward the Th2 cell response (Figure 4E–H). Furthermore, we evaluated fasting glucose levels in mice after STZ treatment. When FBG was stable above 11.1 mmol/L, we considered the modeling to be successful. Intriguingly, compared to the CK and STZ groups, OVA + STZ mice had a higher modeling success rate, which indicated mice in the OVA + STZ group have a higher susceptibility to diabetes (Figure 4I). Overall, these results indicated that when mice were in a state of sensitization, they presented an immune imbalance and showed increased susceptibility to diabetes.

Figure 5A–B. it is worth noting that jejunal villi were damaged most in OVA + STZ group mice (Figure 5A). Furthermore, compared to the CK group, the STZ + OVA and OVA + STZ groups showed a significant increase in Zonulin level in the blood serum, suggesting that intestinal barrier function was severely impaired [52] (Figure 5C). At the molecular level, we found increased expression of ZO-1, claudin-1, and occludin in the jejunum, indicating a self-negative feedback mechanism (Figure 5D). Notably, we observed enhanced expression of claudin-1 and occludin in OVA + STZ mice compared to that in OVA mice, indicating the jejunal barrier was damaged most in the OVA + STZ group (Figure 5D). In addition, we found an increased number of mast cells and an enhanced expression of mMCP-1 in the jejunum after high-dose OVA stimulation, compared to the CK group, indicating that barrier function and homeostasis were destructed [53] (Figure 5E,F). Altogether, these results suggested that when the mice were sensitized by allergens the intestinal barrier function was severely impaired and mast cell activation was promoted.

### 3.5. Food Allergy Affects Diabetes by Demoting the Secretion of GLP-1

To understand the molecular mechanism by which food allergy affects diabetes, we evaluated the secretion of GLP-1. GLP-1 can bind to the GLP-1R to activate insulin secretion in pancreatic β-cells [54]. Therefore, to further evaluate the role of GLP-1 as a positive regulator of intestinal barrier function, we measured GLP-1 and insulin levels in blood serum. A significant decrease was observed in GLP-1 levels in all animal groups compared to the CK group (Figure 6A). Strikingly, GLP-1 levels were significantly reduced in the OVA + STZ mice compared to those in the other groups (Figure 6A). In the pancreas, GLP-1 is now known to induce the augmentation of glucose-stimulated insulin secretion [55]. We found that the OVA + STZ group also showed decreased insulin levels compared to the OVA group (Figure 6B). GLP-1 secretion increased in an IL-6-dependent manner [56]. Interestingly, IL-6 levels were significantly enhanced in the serum and decreased in the jejunum, suggesting that GLP-1 secretion increased in an IL-6-dependent manner (Figure 6C). TNF-α induced systemic inflammation and reduced GLP-1 levels [57]. In our experiment, the TNF-α levels in blood serum showed a significant increase in the OVA, STZ + OVA, and OVA + STZ groups, compared to those in the CK group (Figure 6D). High levels of glucose induced the release of GLP-1 through the sodium–glucose cotransporter 1 (SGLT1) and, to a lesser extent, the glucose transporter 2 (GLUT2). Critically, the expression of SGLT1 and GLUT2 decreased in all cases compared to the CK group, suggesting that GLUT2 inhibited the secretion of glucose-induced GLP-1 (Figure 6E). Notably, we found a significant decrease in GLUT2 in the OVA + STZ group compared to the OVA group (Figure 6E). Overall, these results indicated that when the mice are in the OVA-sensitized state, GLP-1 secretion inhibits, in accompaniment with decreased insulin, IL-6, TNF-α, SGLT1, and GLUT2 levels.

### 3.6. Food Allergy Affects Diabetes by Upregulating the Expression of PI3K/Akt/mTOR/NF-κB p65

Next, we aimed to elucidate the mechanism between food allergy and diabetes. Previous studies have shown that mTOR is a critical regulator of intestinal barrier damage and secretion of GLP-1, and the proposed mechanism of action of GLP-1 is believed to occur through activation of the PI3K/Akt signaling pathway [36,58]. We evaluated the mRNA and protein expression of mTOR and other related immunological and metabolic markers, including PI3K, Akt, and NF-κB p65.

Among those, the PI3K/Akt signaling pathway was activated, and the expression of PI3K/Akt/mTOR/NF-κB p65 showed a significant increase in the mRNA levels compared to the CK group (Figure 7A). At the protein level, there was no significant difference in PI3K expression levels between each group. Compared to the STZ group, the expression of PI3K in the OVA + STZ group enhanced significantly and significantly upregulated the expression levels of mTOR, and NF-κB p65 in STZ + OVA and OVA + STZ mice, compared to other control groups (Figure 7B,C), which indicated that STZ and OVA demoted GLP-1 secretion by up-regulating the expression levels of PI3K, mTOR, and NF-κB p65. Intriguingly, we observed how the OVA + STZ group enhanced protein expression in PI3K and mTOR levels, compared to the STZ group, which may explain why mice in the OVA + STZ group presented increased immune imbalance to Th2 humoral response to and showed increased susceptibility to diabetes when they were in a state of sensitization (Figure 7B,C).

Taken together, these results emphasized that the PI3K/Akt/mTOR/NF-κB p65 pathway, especially mTOR, played an important role in the mechanism by which food allergies affect diabetes.

## 4. Discussion

This study suggests that food allergies have interactions with diabetes. Food allergy increased the risk of STZ-induced diabetes in mice, which promoted food allergen-induced damage of the jejunum barrier and direct uptake of dendritic cells, induced mast cell activation, increased IL-4, IL-5, IL-6, mMCP-1, and TNF-α secretion, reduced GLP-1 secretion in the jejunum, and decreased insulin secretion in the pancreas. (Figure 8).

The quality of life of many patients with food allergies and diabetes is affected. Previous studies have found that T1D, T2D, and DM are associated with allergic disease [18,19,20,21,22,23,24,25]. In addition, a previous study showed a positive correlation between allergic disease and diabetes with an odds ratio of 1.25, indicating that allergic disease is a risk factor for diabetes [59]. Thus, it is important to establish a model and identify a mechanism linking food allergy and diabetes. To explore this relationship, we established a BALB/c animal model to explore the immune and metabolic levels of food allergy and diabetes. We used OVA and STZ to build a food allergy and a diabetes mice model, respectively. It has been observed that BALB/c inbred nude and outbred nude mice receiving the highest STZ dose (240 mg/kg) experienced the lowest rate of complications [60]. In the current study, a diabetic allergic mice model showed that diabetes did not disrupt the immune balance in mice. However, allergic diabetic mice showed a more significant tilt toward Th2 immune response (*p* < 0.05) and a higher success rate of modeling, suggesting that OVA sensitization may cause an immune imbalance and increase the susceptibility to diabetes. The results of a previous population study in the United States also suggested that allergic asthma can affect the severity of diabetes [61], and Thomsen and collaborators found that significant positive genetic correlations were found between asthma and T2D [62]. Moreover, the results of another population study in Canada suggested that the relationship between allergic diseases and diabetes may be related to a Th1/Th2 imbalance [59]. However, there was no difference in the jejunum mRNA expression of IL-4 and IFN-γ in OVA, STZ, STZ + OVA, and OVA + STZ mice in our previous study (Appendix A). Similarly, there was no relation between TH2-mediated atopy and TH1-mediated autoimmune disorders in the American Third National Health and Nutrition Examination Survey, with an odds ratio of 1.01 (95% CI: 0.61, 1.67) [63]. In a classical mice model of autoimmune diabetes, increased Th1 has been found in the mesenteric lymph nodes (MLNs) and pancreatic lymph nodes (PaLNs) [64]. Moreover, evidence supporting this possibility includes enhanced Th1 cytokine expression in diabetic subjects [65]. In summary, to quote Sheikh and collaborators, ‘the Th1/Th2 model, as currently formulated, might be an oversimplification’ [63].

Mounting evidences have suggested that the gut immune system is involved in autoimmune diabetes development. Up to 75% of individuals with diabetes experience gastrointestinal symptoms [66]. Previous studies have demonstrated that stiffness of the intestinal wall, particularly the jejunum wall, increases over time during DM development [67]. The jejunum is a major site of digestion and absorption within the small intestine [68]. An inflammatory state has been confirmed in the intestines of patients with T1D, T2D, and MD, and abnormal intestinal permeability has been found to be the contributing factor [69,70,71]. In diabetes, the jejunum plays a key role in regulating insulin sensitivity [72]. Additionally, altered intestinal epithelial barrier function and composition have been observed in patients with food allergies [5,73]. In the current study, we showed that OVA and STZ treatments in mice led to histopathological damages in the structure of the jejunum and a significant increase in the number of mast cells, in agreement with previous results. Zonulin is a biomarker of intestinal barrier dysfunction [57]. Its increased level in the serum can activate the dense connexin ZO-1, claudin-1, and occludin, which are fundamental for the maintenance, integrity, and normal function of the intestinal mechanical barrier [74]. Mast cells participate in specific and nonspecific immune responses related to inflammatory and metabolic diseases (such as diabetes). Specifically, mast cells contribute to sensitization to food allergens and stimulate the release of histamine, protease, cytokines, and chemokines [34,35,45,75], which induce allergic reactions and diabetes [76,77].

GLP-1 is a hormone secreted by intestinal L cells, and its response to glucose is limited to small intestinal L cells. L cells in the jejunum are capable of secreting high levels of GLP-1 [78]. GLP-1 binds to the GLP-1R, a G-protein–coupled receptor widely expressed in the pancreas, gastrointestinal tracts, kidneys, lungs, and hearts [79]. Glp-1R mRNA expression is the highest in the epithelial fraction of the jejunum, followed by the ileum and colon [80]. Recently, He and collaborators reported that gut intraepithelial T cells regulate GLP-1 bioavailability by capturing GLP-1 on GLP-1Rs and impacting L cell numbers [81]. GLP-1R agonists can regulate OVA-induced airway inflammation and mucus secretion in mice [35], which are related to the inhibitory effect of GLP-1R signaling on Th2 inflammation [82]. Rapid secretion of GLP-1 occurs before the nutrients reach the ileum and colon, where large numbers of L cells are concentrated [83]. GLP-1 secretion is regulated by IL-6 signaling [84], and inflammatory stimuli increase GLP-1 secretion in an IL-6-dependent manner [57]. In contrast with previous studies, our results showed that IL-6 levels in the serum significantly decreased after OVA and STZ stimulation (*p* > 0.05). This discrepancy may be related to the different animal strains used in the study [85], different sampling sites [86], and different models [87]. SGLT1 and GLUT2 play a dominant role in controlling glucose-stimulated GLP-1 release from human L cells [88].

A large number of previous studies have found that intestinal mTOR regulates GLP-1 production in mouse L cells, and mTOR may link the energy supply to GLP-1 production in L cells [38,89,90]. The PI3K/Akt/mTOR signaling pathway is important not only in the development of cancer but also in the proliferation, adhesion, migration, metabolism, and survival of normal cells [91]. The PI3K/Akt/mTOR pathway in innate immune cells inhibits autophagy in allergic asthma [92], and it is considered an emerging therapeutic target [93]. Moreover, a study found that the PI3K/Akt/mTOR signal transduction pathway constitutes the intersection of allergic asthma and cataracts [94]. In a study on diabetes, it was found that the PI3K/Akt/mTOR signaling pathway is associated with the apoptosis of pancreatic β cells in Wistar rats with STZ-induced diabetes [95]. In addition, this pathway could also be involved in post-injury healing and autophagy dysfunction [96,97,98], and its inhibition has been suggested for the treatment of diabetic retinopathy [99]. The transcription factor NF-κB plays an important role in both innate and adaptive immunity [100]. Lipid accumulation in the liver leads to subacute liver “inflammation” through NF-κB activation and downstream cytokine production. This can cause insulin resistance locally in the liver and throughout the body [101].

## 5. Conclusions

Mice models were used to study the interaction between food allergy and diabetes. Pathological section, mast cell staining analysis, glucose and insulin tolerance test, ELISA, real-time PCR, and Western blot helped to explore the potential mechanisms by which food allergy promotes diabetes. The main findings of the paper are as follows:(1)The diabetic allergic (STZ + OVA) model was successfully developed by basic growth status and glucose sensitivity. Compared to the STZ group, STZ treatment could not further disrupt the secretion of immunoglobulin and cytokines induced by OVA administration;(2)Allergic diabetic (OVA + STZ) model was successfully developed by basic growth status and glucose sensitivity. Compared to the OVA group, when mice were in a state of sensitization, the pathological morphology of the viscera, the composition of peripheral blood leukocytes, and the metabolic level were altered, the secretion of immunoglobulin and cytokines could further disrupt and would have a higher susceptibility to diabetes;(3)Food allergy affects diabetes by promoting jejunal barrier destruction through the damage of tight junction proteins and the proliferation and activation of mast cells;(4)Food allergy affects diabetes by demoting the secretion of GLP-1. When mice were in the OVA-sensitized state, secretion of GLP-1 secretion is inhibited, in accompany by decreased insulin, IL-6, TNF-α, SGLT1, and GLUT2 levels;(5)Food allergy affects diabetes by upregulating the expression of PI3K/Akt/mTOR/NF-κB p65, especially mTOR, which plays an important role in the mechanism by which food allergy affects diabetes.

## Figures and Tables

**Figure 1 foods-11-03758-f001:**
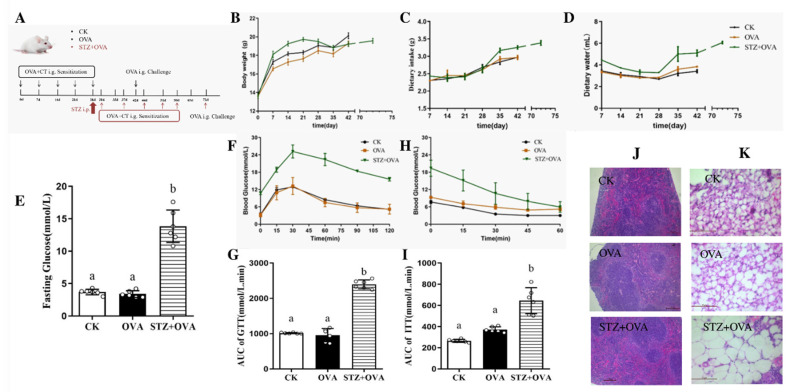
Establishment of diabetic allergy (STZ + OVA) mice model. (**A**), schematic figure of the effect of diabetes on food allergy: mice were divided into CK, OVA, and STZ + OVA group mice. OVA group mice were given OVA and CT on day 0, 7, 14, 21, 28, and 42. The STZ+ OVA group mice were given STZ on day 28. On day 30, 37, 44, 51, 58, and 72, mice were given OVA and CT. i.g., intragastric. (**B**), Body weight. (**C**), Dietary intake. (**D**), Dietary water. (**E**), Fasting glucose. (**F**), Glucose tolerance. (**G**), Area under curve (AUC) of glucose tolerance. (**H**), Insulin tolerance. (**I**), AUC of insulin tolerance. (**J**), H&E staining of spleen (scale bar = 100 µm). (**K**), H&E staining of visceral adipose tissue (scale bar = 50 µm). The error bars indicated the mean ± SEM. *n* = 6. Different lowercase letters in the figure represent significant differences between the groups (*p* < 0.05 by one-way ANOVA followed by SNK).

**Figure 2 foods-11-03758-f002:**
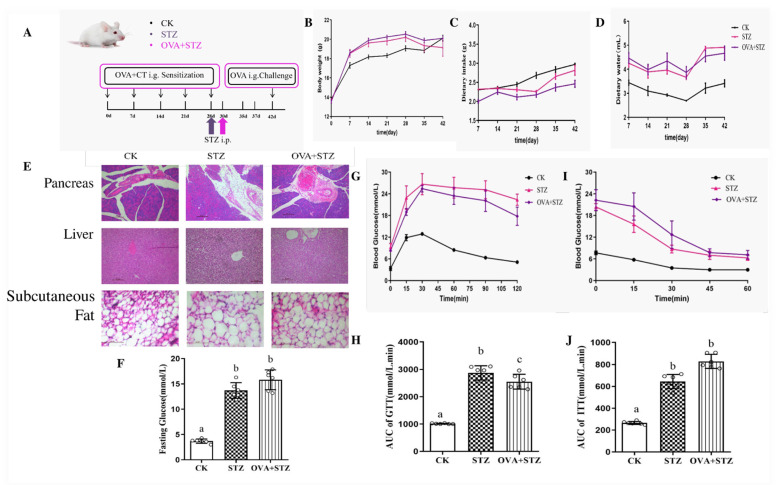
Establishment of allergic diabetic (OVA + STZ) mice model. (**A**), Schematic figure of the effect of food allergy on diabetes: mice were divided into CK, STZ, and OVA + STZ group mice. The STZ group mice were given STZ on day 28. The OVA + STZ group mice were given OVA and CT on day 0, 7, 14, 21, 28, and 42. On day 30, mice were given STZ. i.g., Intragastric. (**B**), Body weight. (**C**), Dietary intake. (**D**), Dietary water. (**E**), H&E staining of pancreas, liver, and subcutaneous fat (scale bar = 50 µm). (**F**), Fasting glucose. (**G**), Glucose tolerance. (**H**), AUC of glucose tolerance. (**I**), Insulin tolerance. (**J**), AUC of insulin tolerance. The error bars indicated the mean ± SEM. *n* = 6. Different lowercase letters in the figure represent significant differences between the groups (*p* < 0.05 by one-way ANOVA followed by SNK).

**Figure 3 foods-11-03758-f003:**
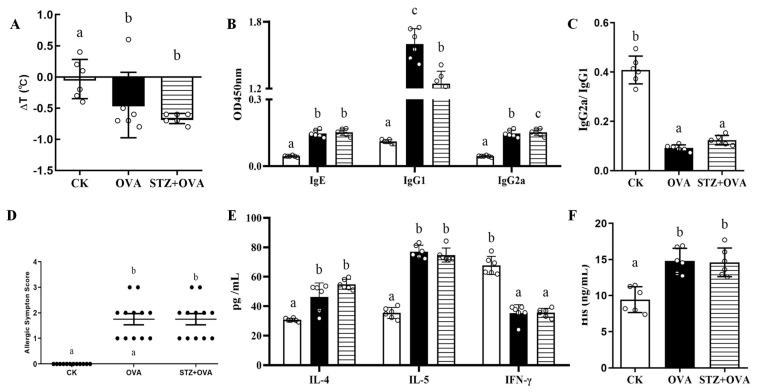
Effects of diabetes on food allergy. (**A**), Rectal temperature change. (**B**), Serum-specific antibody levels of IgE, IgG1, and IgG2a. (**C**), Serum-specific antibody IgG2a/IgG1 level. (**D**), Food allergy symptom score. (**E**), Cytokines levels of IL-4, IL-5, and IFN-γ. (**F**), Histamine levels. The error bars indicated the mean ± SEM. *n* = 6. Different lowercase letters in the figure represent significant differences between the groups (*p* < 0.05 by one-way ANOVA followed by SNK).

**Figure 4 foods-11-03758-f004:**
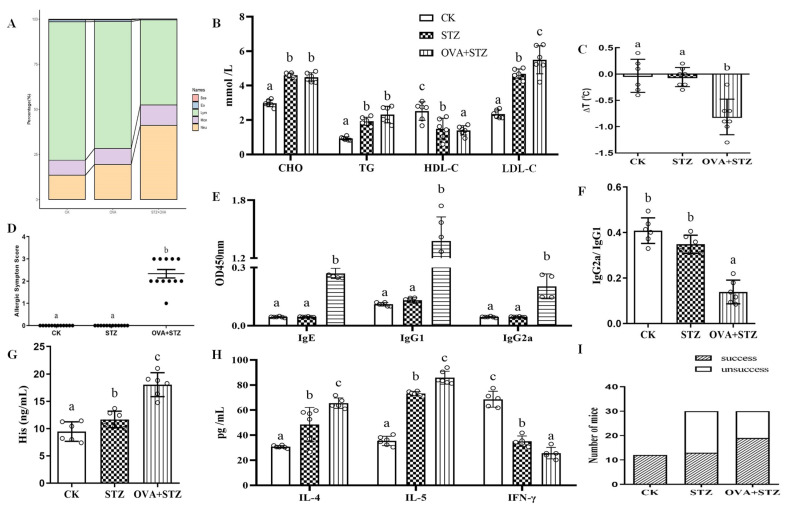
Effects of food allergy on diabetes. (**A**), The proportion of leucocyte (Lymphocytes, monocytes, and granulocytes) component distribution. Orange, purple, green, blue, and red represented neutrophile, monocyte, lymphocyte, eosinophils, and basophilic, respectively. (**B**), Serum levels of CHO, TG, HDL-c, and LDL-c. (**C**), Rectal temperature change. (**D**), Food allergy symptom score. (**E**), Serum-specific antibody levels of IgE, IgG1, and IgG2a. (**F**), Serum specific antibody IgG2a/IgG1 level. (**G**), Histamine levels. (**H**), Cytokines levels of IL-4, IL-5, and IFN-γ. The error bars indicated the mean ± SEM. *n* = 6. Different lowercase letters in the figure represent significant differences between the groups (*p* < 0.05 by one-way ANOVA followed by SNK). (**I**), The success rate of animal experimental models (by the chi-square analyses).

**Figure 5 foods-11-03758-f005:**
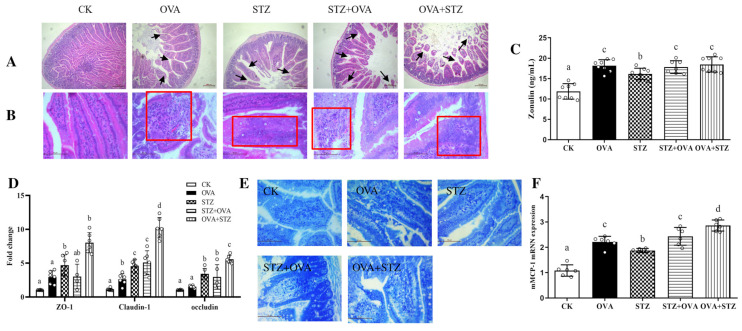
Food allergy affects diabetes by promoting jejunal barrier destruction. (**A**), H&E staining of jejunum (scale bar =100 μm). The arrows in the Histo means significant changes in intestinal villi structure. (**B**), H&E staining of jejunum (scale bar = 50 μm). (**C**), Serum levels of Zonulin. (**D**), mRNA expression of tight junction proteins (ZO-1, Claudin-1, occludin) in jejunum. (**E**), Toluidine blue staining of jejunum (scale bar = 50 μm). (**F**), mRNA expression of mMCP-1 in jejunum. The error bars indicated the mean ± SEM. *n* = 6. Different lowercase letters in the figure represent significant differences between the groups (*p* < 0.05 by one-way ANOVA followed by SNK).

**Figure 6 foods-11-03758-f006:**
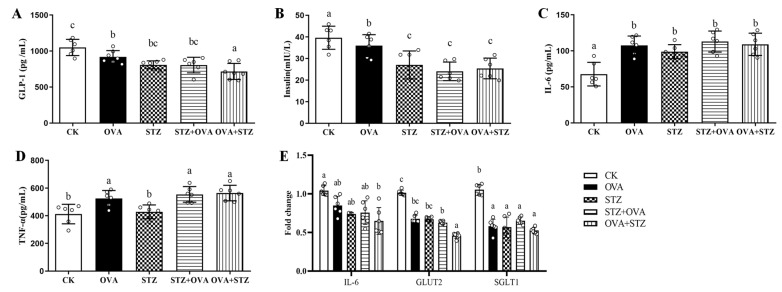
Food allergy affects diabetes by demoting the secretion of GLP-1. (**A**), Serum levels of GLP-1. (**B**), Serum levels of insulin. (**C**), Serum levels of IL-6. (**D**), Serum levels of TNF-α. (**E**), mRNA expression of IL-6, GLUT2, and SGLT1 in jejunum. The error bars indicated the mean ± SEM. *n* = 6. Different lowercase letters in the figure represent significant differences between the groups (*p* < 0.05 by one-way ANOVA followed by SNK).

**Figure 7 foods-11-03758-f007:**
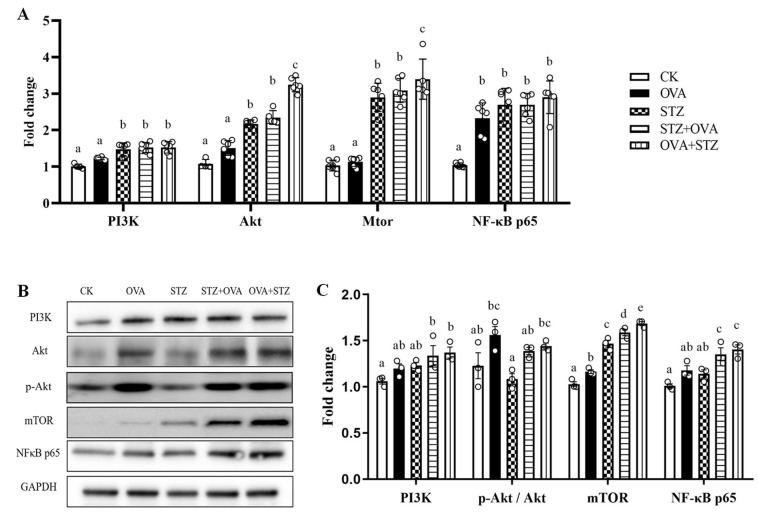
Food allergy affects diabetes by upregulating the expression of PI3K/Akt/mTOR/NF-κB p65. (**A**), mRNA expression of PI3K/Akt/mTOR/NF-κB p65 in jejunum. (**B**), Western blotting of PI3K/Akt/mTOR/NF-κB p65 in jejunum. (**C**), Quantitative analysis of Western blot results of Figure 3M. The error bars indicated the mean ± SEM. *n* = 6. Different lowercase letters in the figure represent significant differences between the groups (*p* < 0.05 by one-way ANOVA followed by SNK).

**Figure 8 foods-11-03758-f008:**
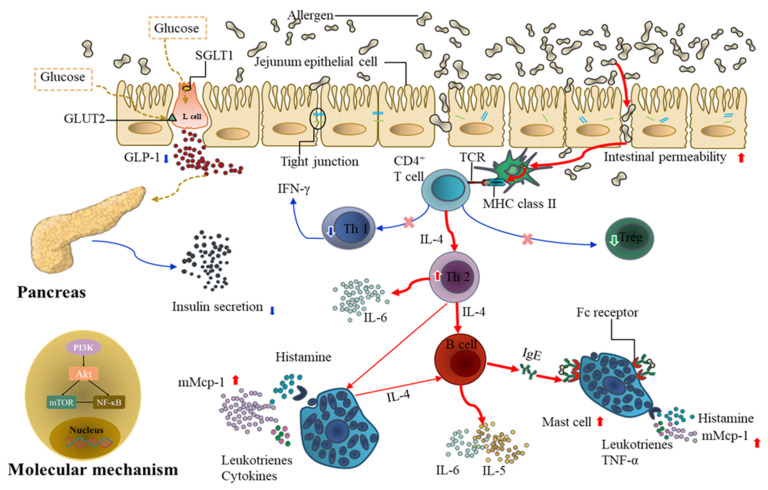
A model of the potential mechanism underlying the food allergy on diabetes.

**Table 1 foods-11-03758-t001:** List of primer pairs used for real-time quantitative PCR.

Gene Name	Forward (5′-3′)	Reverse (5′-3′)
β-actin	GATTACTGCTCTGGCTCCTAGC	GACTCATCGTACTCCTGCTTGC
ZO-1	TTTTTGACAGGGGGAGTGG	TGCTGCAGAGGTCAAAGTTCAAG
Claudin-1	CGGGTTGCCTGCAAAGT	ATGTCCGGCCGATGCTCTC
occludin	CTTTGGCTGCTGTTGGGTCTG	AGCCAGGAGCCTCGCCCCGCAGCTGCA
mMCP-1	CAGATGTGGTGGGTTTCTCA	GCTCACATCATGAGCTCCAA
IL-6	CGTGGAAATGAGAAAAGAGTTGTGC	ATGCTTAGGCATAACGCACTAGGT
SGLT1	CGGAAGAAGGCATCTGAGAA	AATCAGCACGAGGATGAACA
GLUT2	TCTTCACGGCTGTCTCGTG	AATCATCCCGGTTAGGAACA
mTOR	ACCGGCACACATTTGAAGAAG	CTCGTTGAGGATCAGCAAGG
PI3K	CACTCAGCCCATCTATTTCCAG	TCTTGGATCTTCACCTTCAGC
AKT	GACTGACACCAGGTATTTCGATGA	CTCCGCTCACTGTCCACACA
NF-κB p65	GCATTCTGACCTTGCCTAT	ACCGCCACTACCGAACAT

## Data Availability

All the data regarding this work are presented in this manuscript.

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
