# Peer review of "The Interaction of Food Allergy and Diabetes: Food Allergy Effects on Diabetic Mice by Intestinal Barrier Destruction and Glucagon-like Peptide 1 Reduction in Jejunum"

_foods, 2022, doi:10.3390/foods11233758_

Round 1
Author Response
Dear reviewer,
Thank you very much for every suggestion, which is very helpful for me to revise.
I recently received major revisions with respect to our submitted manuscript “The Interaction of Food Allergy and Diabetes: Food Allergy Effects on Diabetic Mice by Intestinal Barrier Destruction and Glucagon-like Peptide 1 Reduction in Jejunum” (Manuscript ID: 1983412) from your esteemed journal, Foods.
Hereto, we have completed the revised manuscript and the point-to-point comments. I am willing to wait an extra round of external review to improve the quality of our paper further. In the response to reviewers, comments were marked in black, corresponding answers were marked in blue and the amendments in the revised manuscript were highlighted in yellow to make the expression clearer. Furthermore, we used “Track Changes” function in our manuscript.
Thank you very much for giving us the precious chance to revise our manuscript again. If there are any questions regarding this revised manuscript, please do not hesitate to contact me.
Best Regards,
Yanjun Gu & Huilian Che

Reviewer 2 Report
The authors tried to address the relevance of the food allergy in the development of the DM. They described several animal models that mimicked the disease development and they included all the relevant controls. The way the authors write the manuscript is simple and clear, trying to make it easier to the readers. However the description of the challenge should be described with more detailed and the figures need to be strongly improved. Also is necessary that the authors specify the statistical test employed for every single experiment; in some figures the statistics are missing.
Regarding to the immunological part related with the food allergic, I would recommend that the authors performed a deeper analysis for the Th2 populations in the gut. It will give them a better understanding about what is happening in the small intestine after the sensitization alone or with the STZ.
Below the authors could find some of the changes that should be made to improve the manuscript:
-Line 147, specify the challenge method more detailed.
-On line 153, when the authors describe the s allergic symptoms, they are missing the diarrhea as a crucial one to score the animals symptoms
-Line 188 is not clear what the authors meant with “after the large OVA stimulation for 45 min”. Please be more specific in the experimental procedure.
-For the statistical analysis in line 223 the authors should specify better the statistical analysis the use for each experiment. This should be applied to all the figures of the manuscript.
-For the results in the establishment of diabetic allergic (STZ+OVA) mice models, the authors did not show the STZ group alone. I would recommend it in order to see the effect alone of the STZ and with the OVA.
-Line 263, when the authors said “during the whole 30 mins” is not clear how they proceed.
-Line 269, there is a mistake when the authors named the figure 2. They wrote (Fig. 12B-C, E- F). “
-Lines 401, 405 and 410 are not referring to the correct figure. It should be figure 7
Author Response

(The authors gave the same response as above.)
